# Three-Dimensional Comparison of the Maxillary Surfaces through ICP-Type Algorithm: Accuracy Evaluation of CAD/CAM Technologies in Orthognathic Surgery

**DOI:** 10.3390/ijerph191811834

**Published:** 2022-09-19

**Authors:** Andrea Cassoni, Luigi Manganiello, Giorgio Barbera, Paolo Priore, Maria Teresa Fadda, Resi Pucci, Valentino Valentini

**Affiliations:** 1Department of Oral and Maxillofacial Sciences, Sapienza University of Rome, Via Caserta 6, 00161 Rome, Italy; 2Oncological and Reconstructive Maxillo—Facial Surgery Unit, Policlinico Umberto I, Viale del Policlinico 155, 00161 Rome, Italy

**Keywords:** ICP algorithm, orthognathic surgery, virtual surgical planning, CAD/CAM, splint, patient-specific implant

## Abstract

Purpose: This retrospective study aims to compare the accuracy of two different CAD/CAM systems in orthognathic surgery. The novelty of this work lies in the method of evaluating the accuracy, i.e., using an Iterative Closest Point (ICP) algorithm, which matches a pair of 2D or 3D point clouds with unknown dependencies of the transition from scan *s*_(k)_ to scan *s*_(k+1)_. Methods: The study population was composed of ten patients who presented to the Maxillofacial Surgery Department of the University “Sapienza” of Rome for the evaluation and management of skeletal malocclusions. The patients were divided into two groups, depending on the technique used: group 1: splintless group (custom-made cutting guide and plates); group 2: splint group (using a 3D-printed splint). STL files were imported into Geomagic^®^ Control X™ software, which allows for comparison and analysis using an ICP algorithm. The RMSE parameter (3D error) was used to calculate the accuracy. In addition, data were compared in two different patient subgroups. The first subgroup only underwent a monobloc Le Fort I osteotomy (*p*-value = 0.02), and the second subgroup underwent a Le Fort I osteotomy associated with a segmental osteotomy of the maxilla (*p*-value = 0.23). Results: Group 1 showed a 3D error of 1.22 mm ± SD 0.456, while group 2 showed a 3D error of 1.63 mm ± SD 0.303. These results have allowed us to compare the accuracy of the two CAD/CAM systems (*p*-value = 0.09). Conclusions: The ICP algorithm provided a reproducible method of comparison. The splintless method would seem more accurate (*p*-value = 0.02) in transferring the surgical programming into the operating room when only a Le Fort I osteotomy is to be performed.

## 1. Introduction

Dentofacial dysmorphosis is characterized by retrognathism, prognathism, and asymmetry. The occlusal relationship between maxillary and mandibular teeth is a fundamental consideration in complete mouth restoration. Angle proposed a classification of malocclusion that is still relevant today: he suggested that normal occlusion was based fundamentally on the position of the permanent first molars. If these teeth were in the correct relationship and the remaining teeth occupied a smoothly curved line of occlusion, a normal occlusion would result [1,2]. As a result, Angle’s classification has been widely used in orthognathic surgery as part of the initial diagnosis and treatment goals.

Orthognathic surgery to reposition the maxilla, mandible, and chin dramatically enhances facial balance and proportion. Orthognathic surgery serves two unrelenting masters, the soft tissue and the skeleton, with the functional and aesthetic goals of achieving level and class I dental occlusion, facial balance, and proportion. Current concepts further build on improved technical proficiency and a perioperative safety profile, superior bone fixation materials, and a better understanding of bone-healing and soft-tissue responses. Current controversies include the sequence of treatment (maxilla first vs. mandible first vs. surgery first), the traditional stone model vs. modern virtual surgical planning, and adjunct procedures that enhance bony movements [3].

Over time, orthognathic surgery has taken advantage of technological evolution. Nowadays, it is possible to perform computer-assisted surgical programming and transfer the planning to the operating room through devices designed and manufactured with computer assistance (CAD/CAM technologies) [4,5]. To date, for the diagnosis of dentofacial dysmorphosis, we have been relying almost entirely on reference points, planes, and angles, all landmarks identified on cephalometric images. This diagnostic procedure is not only time-consuming but also greatly influenced by the practitioner’s skill level [6].

New methods have been developed to determine the accuracy of CAD/CAM devices and to evaluate and compare the improvements they offer in surgery [7].

This retrospective study aims to evaluate, by 3D comparison, the accuracy and effectiveness of two different CAD/CAM systems available on the market—splints and custom-made cutting guides and plates—in reproducing the movement designed with virtual surgical planning (VSP). The novelty of this work lies in the method of evaluating the accuracy, i.e., using an Iterative Closest Point (ICP) algorithm, which matches a pair of 2D or 3D point clouds with unknown dependencies of the transition from scan *s*_(k)_ to scan *s*_(k+1)_ [8]. In this way, we can study the accuracy of the postoperative results of orthognathic surgery without relying on reference points, planes, or angles.

The 3D comparison was performed using a piece of software that uses an ICP-type algorithm (Geomagic^®^ Control X™ 2018—Rock Hill, SC, USA). This algorithm is commonly used to assess 3D surface changes by measuring point-to-point distances between surfaces [9]. The program automatically recognizes STL (Standard Triangulation Language) surfaces (VSP 3D model and 3D model obtained from postoperative TC), allowing for the analysis of the measurements and deviation between the two models without the need to identify any cephalometric landmarks [10].

## 2. Materials and Methods

### 2.1. Selection of the Sample

The study population was composed of ten patients who presented, between November 2018 and November 2019, to the Maxillofacial Surgery Department of the University “Sapienza” of Rome for the evaluation and management of skeletal malocclusions. Inclusion criteria: patients with class II or III malocclusion according to Angle’s classification; patients with condylar hyperplasia; patients who underwent pre- and postoperative CBCT (Cone Beam CT) and a VSP simulated by Dolphin Imaging. Exclusion criteria: patients with syndromic craniofacial disorders; patients only requiring surgery to the mandibular bone. The patients were divided into two groups: group 1 was treated with a splintless technique, and group 2 with a splint technique. In addition, each group was divided into two subgroups, depending on whether a segmental osteotomy of the maxilla was performed (Figure 1 and Appendix A). The study was approved by the Research Ethics Committee of the University “Sapienza” of Rome.

### 2.2. Images Acquisition

CBCT was acquired before and one month after surgery with a 3D Scanorax (Scanora^®^ 3DX cone Beam CT–Soredex, Inc., Milwaukee, WI, USA) set with axial cuts of 0.3 mm and FOV of 180 × 165 mm (90 kV, 10 mA). The study was conducted according to the following protocol: the patient seated with the head in its natural position (NHP) stabilized by a head support and the mandible in maximum intercuspation [11]. Informed patient consent regarding the possible use of the clinical and radiographic documentation was signed.

### 2.3. 3D Virtual Surgical Programming (VSP)

The treatment plan is expressed graphically through VSP Dolphin Imaging, which allows one to visualize the multidimensional correction at the dental level, quantize skeletal displacements, and accurately transfer the planning into the operating room. VSP represents a supporting pillar in the evolutionary path of orthognathic surgery [12,13].

VSP started with the acquisition of a patient’s preoperative CBCT DICOM and STL files, obtained by scanning plaster models of dental arches set in the final position. These data were imported into the surgical programming software: Dolphin Imaging 11.9 Premium (Dolphin Imaging & Management Solutions, Chatsworth, CA, USA). The DICOM files were used to create a 3D model of the patient’s skull, and they were also merged with the STL file of the plaster model of the alveolar ridges. Then, from the resulting 3D model, the software proceeded to reconstruct the 2D X-rays. After thresholding, the model was transformed from a DICOM file to an STL file. Ricketts comprehensive 3D cephalometric analysis was performed. The piggyback function automatically moved the skeletal bases into the planned position, in which the plaster models had been scanned. Then, the maxilla and mandible, keeping the occlusion fixed, were moved together to find the best skeletal position. At this point, two different paths could be followed: in the splintless surgery, the surgical plan was transferred to an external company, and the surgeon discussed it via web meetings with the engineers. Under surgeons’ and orthodontists’ guidance, the engineers would design and print the cutting guide and the patient-specific titanium plates. In the splint surgery, once the intermediate and final splints were drawn using the Dolphin software, the STL files for the splints were exported and sent to an in-house 3D printer.

### 2.4. Surgical Techniques

We adopted a maxillary-first surgical sequence in all ten cases, performing Le Fort I osteotomies in one piece or in a segmental fashion, according to the VSP. The first group, comprising five patients, was treated with splintless surgery using patient-specific implants (PSIs). The PSI system was composed of a cutting guide to perform precise osteotomies and pre-drill the bone, and custom-made plates, used to fix the maxilla once the maxillary osteotomy and pre-drilled holes had been realized. The three-dimensional shape of these devices already contained the movement information so that the maxilla would automatically move into position. In the second group, also composed of five patients, Le Fort I osteotomies were performed freehand by measuring the amount of bone to be removed with a divider caliper. The maxilla was moved in its final position by blocking it on the mandible with an intermediate splint. The maxillary–mandibular complex was rotated upward to allow bony contact between the mobilized maxillary stump and its fixed counterpart and fixation employing stock plates. Then, the intermediate splint was removed, and the bilateral sagittal split osteotomy was performed. All osteotomies were performed with piezosurgery.

### 2.5. Processing of Data

To evaluate the accuracy of the two CAD/CAM systems, all patients underwent postoperative CBCT. A 3D virtual model was created and converted into an STL file related to the virtual surgical programming (T0) and postoperative CBCT (T1) (Figure 2). The two STL files were imported into Geomagic^®^ Control X™ 2018 software to evaluate, with an ICP algorithm, the differences in terms of the maxillary position between T0 and T1 (ΔT). The mesh of the model reconstructed by postoperative CBCT (T1) and the mesh derived from VSP (T0) were imported into the same three-dimensional space (Figure 3). For the next alignment phase (bottom image of Figure 3), two regions of interest (ROIs) were selected on the T0 model. The first region of interest (ROI1) corresponded to the orbital frames and the frontal and zygomatic bones. The second region (ROI2) corresponded to the maxilla in the area between the Le Fort I osteotomy and the neck of the teeth. We used ROI1 as a reference, as it was not of interest in surgery, to evaluate the deviation error in the movement of ROI2. The meshes were aligned (initial alignment) and then fitted (optimized alignment) at ROI1 (Figure 4A,B). Then, ROI2 3D analysis was performed (Figure 5). The resulting data showed the discrepancy in positioning with a colormap, the maximum and the minimum deviation points, mean deviation, and root mean square (RMS)—the square root of the mean square (the arithmetic mean of the squares of a set of numbers). The RMS, used as an accuracy parameter (3D error, ΔT), was considered clinically acceptable in a range of values between ±2 mm, as described in the literature by many authors [10]. The colormap was obtained by applying a variable color gradient to the detected distance (Figure 5). To minimize the risk of error, the data were obtained by three different operators using the same PC and software. The data used in the end corresponded to the average of the results of each operator.

### 2.6. Statistical Analysis

All data were analyzed using SPSS 25.0 (SPSS Inc., Chicago, IL, USA). The Kolmogorov–Smirnov test was used to test the normality of the variables. Student’s *t*-test for coupled samples allowed the calculation of the difference between the virtual simulation and the current position of the maxilla. The level of statistical significance adopted was 5% (*p*-value < 0.05). A clinically insignificant margin of error was established, corresponding to ±2 mm in difference from the preoperative program, as used in the literature [14,15,16].

## 3. Results

Ten patients with an average age of 29.7 years (five males with an average age of 31.2 ± 12.2 years and five females with an average age of 28.2 ± 6 years) were selected for this study. All patients underwent bimaxillary surgery. Three patients, affected by condylar hyperplasia, underwent a condylectomy. Double maxillary segmentation was necessary for four patients: two patients were treated with PSIs and two with splints (Wassmund or Schuchardt Osteotomy). Virtual 3D surgical programming was successfully transferred to the operating room. The statistical analysis showed a 3D error of 1.22 mm (SD 0.46) for the first group and 1.63 mm (SD 0.3) for the second. We compared the accuracy of the two CAD/CAM systems (*p*-value = 0.09) (Figure 6 and Appendix A). In addition, the data on accuracy were analyzed in two different patient subgroups:

Patients operated on with monobloc maxillary osteotomy: splint technique vs. splintless technique (*p*-value = 0.02);Patients operated on with segmental maxillary osteotomy: splint technique vs. splintless technique (*p*-value = 0.23).The results are summarized in Figure 1 and Figure 6 and Appendix A.

## 4. Discussion

This study aimed to establish which CAD/CAM technique was more accurate in transferring the surgical program into the operating room. The evaluation of the results used a program that, through an ICP algorithm, compared the positioning of the maxilla obtained by the surgeons with the one planned virtually (3D error, ΔT). The study highlighted the benefits that can be obtained in orthognathic surgery using 3D VSP. The analysis of the results allows us to state that, in general, both the splint and the splintless techniques show clinically acceptable accuracy (<2 mm), in line with the literature [10]. The novelty of this work lies in the method of evaluating the accuracy, i.e., using an Iterative Closest Point (ICP) algorithm, which matches a pair of 2D or 3D point clouds with unknown dependencies of the transition from scan s(k) to scan s(k + 1) [8]. In this way, we can study the accuracy of the postoperative results of orthognathic surgery without relying on reference points, planes, or angles. Computer-assisted programming greatly facilitates the surgical procedure, enabling the surgeon to perform a less complicated and more precise repositioning of the maxilla [12]. The time taken by the operator to perform the entire planning on Dolphin Imaging was found to be between 60 and 120 min. The method using PSIs would seem more accurate in transferring the surgical planning into the operating room in the case of a monobloc Le Fort I osteotomy (*p*-value = 0.02) [10,17]. On the other hand, the comparison is not statistically significant when considering segmental osteotomies (*p*-value = 0.09 and *p*-value = 0.23). The reason could be linked to the presence of anatomical elements that, in segmental surgery, do not allow the extent of skeletal displacement that had been planned. We recommend a study involving a larger cohort to support this hypothesis.

The use of a surgical splint to correct the position of the jaw does not represent an innovation in orthognathic surgery; the real evolution is in how the splint is designed and produced, no longer by hand but by 3D-printing an STL file generated by software [18]. However, the criticalities associated with using the splint in repositioning the jaw persist even with 3D printing: during repositioning, the maxillary bone is bound to the mandible, and this does not give the surgeon, when fixing with plates and screws, complete control of the position of the jaw itself in the three planes of space; therefore, any erroneous traction of the mandible could alter the maxilla’s final position [19]. It follows that splintless surgical treatment is the best procedure, as the innovative execution of an osteotomy with cutting guides and the subsequent fixing with custom-made plates make it possible to overcome the limits of splinting: the maxilla is not attached to the mandible, and thus the latter will not interfere with the final position of the former.

The literature is lacking in high-quality clinical studies using a validated standard method to compare the results obtained and the VSP. There is also a lack of consensus between different authors on the methods of evaluation and validation [7,20]. A standardized 3D analysis is needed to identify further advantages and disadvantages related to the different methods of transferring surgical plans into the operating room [21]. In the literature, different methodologies have been described to compare the three-dimensional changes in hard tissues after orthognathic surgery, but most of these works foresee a human error linked to the positioning of surface landmarks [9,22]. The most used method involves linear and angular measurements. Errors of 2.47 mm on average were reported, even when the landmarks were positioned by expert operators [23]. To reduce the intrinsic error in identifying the reference points, Baan et al. suggested an accuracy analysis method that involves positioning some validated cephalometric landmarks [10]. In the present preliminary study, due to the reasons mentioned above, we decided to use an Iterative Closest Point (ICP) algorithm that allows for the 3D comparison of STL surfaces without the need to identify any cephalometric landmarks, resulting in a 3D congruence analysis free from human error and a visual quantification of the obtained results with colormaps. However, this method has limitations [7]. For example, the necessity of an additional step in mesh rendering, considered a potential source of error, makes some authors prefer voxel-based measurements [24,25]. Moreover, the ICP algorithm can quantify how strongly the obtained data deviate from the reference data; it does not, however, analyze the rotation/translation around the Cartesian axes (*x*, *y*, and *z*) of the repositioned skeletal segments; therefore, it does not show in which direction the error occurred [24,26,27,28]. Other factors that negatively affect the homogeneity of the sample and the assessment are the manual selection of ROI [14], CBTC resolution limits [29], overlay due to plates on the obtained postoperative CBCT mesh, the presence of metal in the area with subsequent surface rendering errors, and finally, the skills possessed by the operator [5,30]. The entire digitalized orthognathic surgical plan can be accurately transferred into the operative room by creating devices, such as splints or custom-made cutting guides and plates, with a computer-aided design and manufacturing process (CAD/CAM) [12]. Surgical splints are not an innovation in orthognathic surgery; the real evolution is in their design and production, as 3D printing allows the production of a wafer perfectly adherent to the surgical plan [18]. Still, issues related to splints also persist with 3D printing. These devices move the maxilla in its final sagittal and transverse positions using the mandible as a fixed base; the maxilla is then moved along the vertical plane by rotating the mandible. This means that an error affecting the maxillary repositioning in all three spatial planes can be caused by the surgeon just by applying too much force on the mandible, forcing the condyle into a non-physiological position [19]. Hence the need to create something that allows the independent repositioning of the maxilla, which would overcome the problems related to splints [10,31]. As splintless surgery has significantly higher costs than the older splint surgery, we wanted to evaluate whether this new method is more accurate and dependable, which would justify its use within the Italian Health Service, which is entirely public and universal. Our data suggest that the splintless method has a higher accuracy rate for monobloc Le Fort I osteotomies (*p*-value = 0.02) [10,17], whereas the comparison between the two methods is not statistically significant for segmental Le Fort I osteotomies (*p*-value = 0.23).

Regarding the limitations of this work, it could be argued that it would be desirable to have a larger sample. This is a retrospective study; therefore, the limits are related to the sample size and methodology. In the future, prospective studies are needed to increase the included subjects and compare the results of the ICP with those of the classical two-dimensional methods.

## 5. Conclusions

This study shows that the ICP algorithm in the Geomagic software (Geomagic^®^ Control X™ 2018—Rock Hill, SC, USA) provides a reproducible method of comparison and alignment between 3D models. A larger cohort of patients is desirable to confirm the results of the study and to identify which is the best technique. In the coming years, better algorithms and new, fully automated methods of 3D comparison will probably be developed, making this kind of surgery even more precise and dependable. The splint and splintless methods both have clinically acceptable accuracy (<2 mm), as reported by other authors [5]. However, the splintless method is more accurate in transferring the surgical plan into the operating room when performing monobloc Le Fort I osteotomies.

## Figures and Tables

**Figure 1 ijerph-19-11834-f001:**
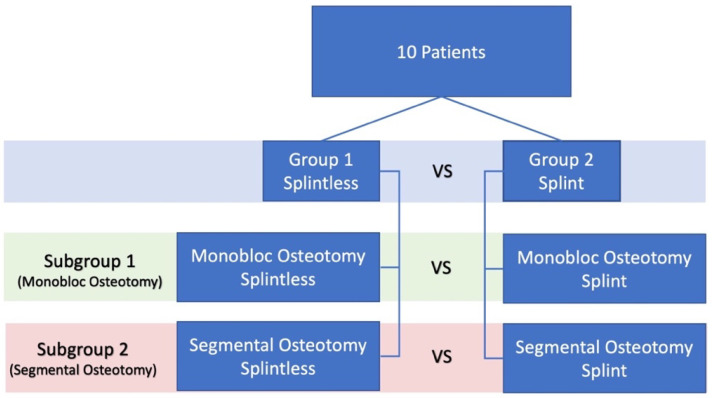
Sample subdivision.

**Figure 2 ijerph-19-11834-f002:**
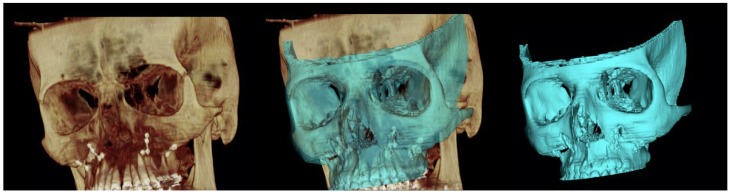
Creation of the STL file starting from the postoperative CBCT.

**Figure 3 ijerph-19-11834-f003:**
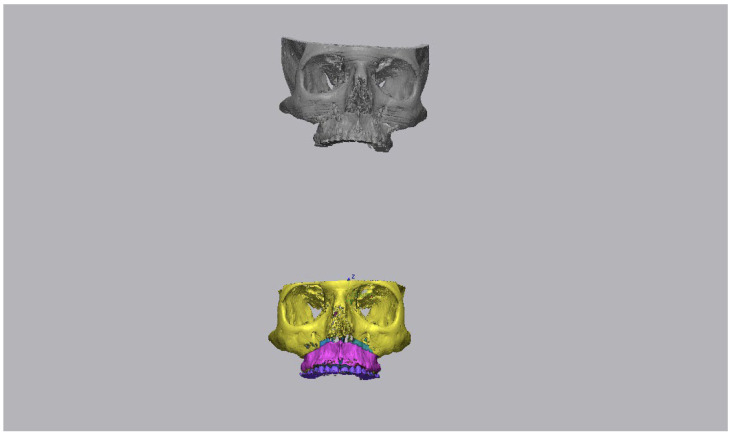
Import of the STL files, obtained, respectively, from the postoperative CBCT and the VSP, within the interface of Geomagic software.

**Figure 4 ijerph-19-11834-f004:**
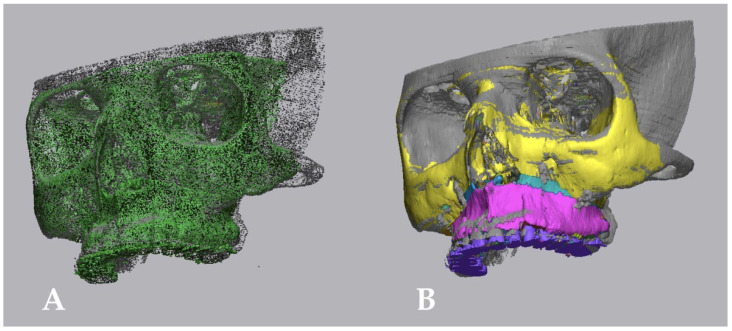
(**A**): Initial alignment of the meshes in ROI1. (**B**): Optimized alignment of the meshes in ROI1.

**Figure 5 ijerph-19-11834-f005:**
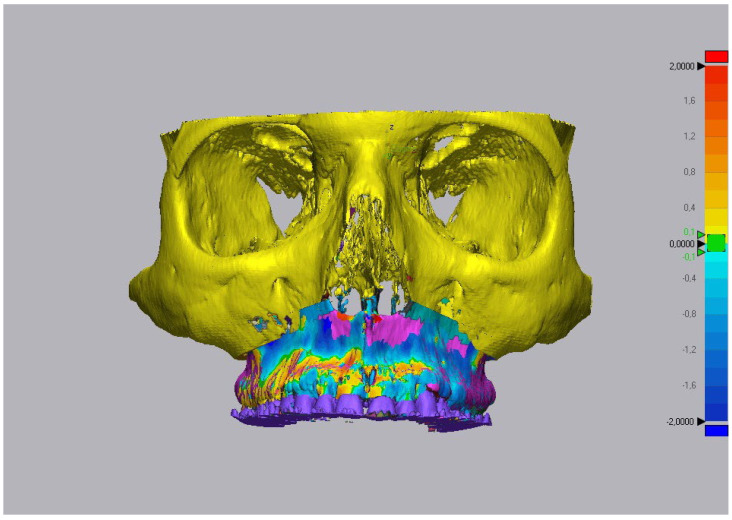
Three-dimensional comparison of the meshes in ROI2, highlighted by colormap.

**Figure 6 ijerph-19-11834-f006:**
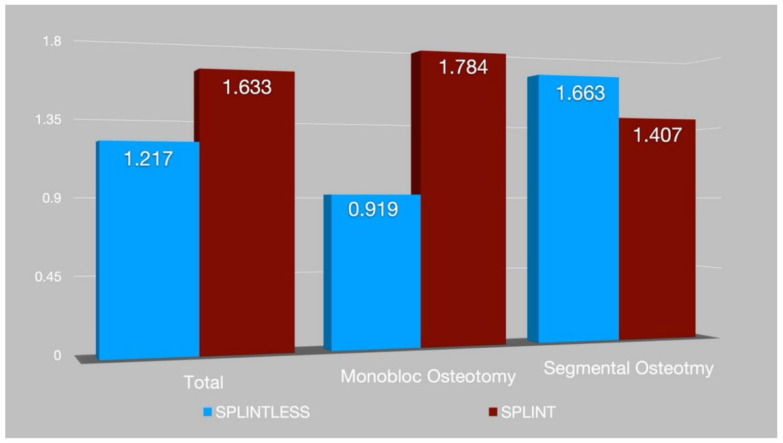
Histogram showing 3D comparison (RMSE) in the three groups analyzed.

## Data Availability

The data presented in this study are available on request from the corresponding author.

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
