# Peer review of "Three-Dimensional Comparison of the Maxillary Surfaces through ICP-Type Algorithm: Accuracy Evaluation of CAD/CAM Technologies in Orthognathic Surgery"

_ijerph, 2022, doi:10.3390/ijerph191811834_

Round 1

Reviewer 1 Report

Dear Author,

The manuscript needs significant improvements. As the subject of the manuscript seems interesting, but it not well-written, as in most instances certian procedures and terminology suddenly appear in the manuscript without proper emphasis on it. I suggest the authors exhaustively edit the manuscript and resubmit. Main points that need improvement: 1. The novelty of this work should be clearly presented mainly, in the abstract. 2. The results and discussion section is uncompleted and confused. Thus, it cannot be easily followed by the readers. The obtained results mainly those regarding the CAD/CAM software should be fully discussed comparing to other studies in the literature. 3. In general, all manuscript was written as a report and not as a scientific article.

Following are some of the detailed points that the author need to emphasize and also might help to improvise the manuscript.

1.In abstracts, please clearly define the groups in the methodology section.  Results directly present with the groups for which the description is not provided in the methodology.  SD should be presented with ±. Line 20 should be before the presentation of results. Line 21; First operated with- please clarify is it Group I. Line 23 with respect to conclusion is not more precise.

2. In introduction, the background is very minimal. The introduction lacks with the information such as need for the study, citation of current evidence, difficulities faced during orthognatic surgery and more importantly the gap in the knowledge; like why this study was needed with its justification. Line 41-42 its not clear to me as what exactly the author is trying to convey.

3. In methodology: It is very much confusion with respect to randomization of samples, procedures employed and group allocation.  The total sample size was 10 but as per the procedural descript the sample size is 11.  Line 51, orthodontic classification should be properly described as the criteria of angle's malocclusion.  Authors need to explain the importance of using VSP Dolphin Imaging. Exclusion criteria is also confusing as  it is mentioned that those patients who already underwent orthognathic surgery were excluded but the title of the study mentions that this investigation was done in orthognathic surgery.  

Results: Line 58, Table 1 I dont find this table 1 and Table 2 in the manuscript.

Line 86: Recruitment of patients is confusing; 2 groups to 2 different methods state clearly.

Line 99: BSSO should be written in fullform

Discussion: It seems the results are not properly discussed to arrive on a conclusion for the study.

Author Response

Dear Editor,

We would like to thank the reviewers for carefully reviewing our manuscript. We have revised the manuscript, accordingly, based on the suggestions. The following are our point-by-point responses. Revisions in the manuscript are shown highlighted in yellow.

An extensive modification of the English language and style was made.

New bibliographic references have been added. [1,2,3,6,8,12,13]

1.In abstracts, please clearly define the groups in the methodology section.  Results directly present with the groups for which the description is not provided in the methodology.  SD should be presented with ±. Line 20 should be before the presentation of results. Line 21; First operated with- please clarify is it Group I. Line 23 with respect to conclusion is not more precise.

  • Figure 1 has been inserted to better explain the division of the sample.
  • ± symbol has been inserted in the standard deviation.
  • Line 20: has been moved to methods.
  • Line 21: inserted reference to subgroup 1.
  • Line 23: has been corrected.

  1. In introduction, the background is very minimal. The introduction lacks with the information such as need for the study, citation of current evidence, difficulties faced during orthognathic surgery and more importantly the gap in the knowledge; like why this study was needed with its justification. Line 41-42 it’s not clear to me as what exactly the author is trying to convey.
  • The introduction has been modified and expanded to include more information for the reader; now it also includes what novelty the study brings and its importance. There are no other studies in the literature that use this same method of analysis and comparison the accuracy of the CAD/CAM vs conventional technique.
  • Line 41 42: has been corrected.

  1. In methodology: It is very much confusion with respect to randomization of samples, procedures employed and group allocation.  The total sample size was 10 but as per the procedural descript the sample size is 11.  Line 51, orthodontic classification should be properly described as the criteria of angle's malocclusion.  Authors need to explain the importance of using VSP Dolphin Imaging. Exclusion criteria is also confusing as it is mentioned that those patients who already underwent orthognathic surgery were excluded but the title of the study mentions that this investigation was done in orthognathic surgery.  

  • Added in figure 1 the flowchart that better explains the sample; corrected the sample size.
  • Line 51: added angle classification.
  • Authors need to explain the importance of using VSP Dolphin Imaging: done.
  • Exclusion criteria: has been corrected.

Results: Line 58, Table 1 I dont find this table 1 and Table 2 in the manuscript.

  • Missing tables have been inserted.

Line 86: Recruitment of patients is confusing; 2 groups to 2 different methods state clearly.

  • Patient recruitment was corrected; see the fig. 1 and supplemental table 1.

Line 99: BSSO should be written in fullform

  • Done

Discussion: It seems the results are not properly discussed to arrive on a conclusion for the study.

  • The Discussion section has been expanded to better explain the results achieved and the usefulness of this work.

Thank you for the review of our paper, and we appreciate the critiques and comments. I am pleased to be available for any further clarification and needs.                                             

Best regards

Reviewer 2 Report

This is a very interesing paper about 3D-comparison of the maxillary surfaces through ICP-type algorithm. Paper is well written. Methods and results are good. I suggest to add just few lines about "limitations of the paper" at the end of discussion section.

Author Response

Dear Editor,

We would like to thank the reviewers for carefully reviewing our manuscript. We have revised the manuscript, accordingly, based on the suggestions. The following are our point-by-point responses. Revisions in the manuscript are shown highlighted in yellow.

An extensive modification of the English language and style was made.

New bibliographic references have been added. [1,2,3,6,8,12,13]

This is a very interesing paper about 3D-comparison of the maxillary surfaces through ICP-type algorithm. Paper is well written. Methods and results are good. I suggest to add just few lines about "limitations of the paper" at the end of discussion section.

Added "limitations of the paper" at the end of the discussion section

Thank you for the review of our paper, and we appreciate the critiques and comments. I am pleased to be available for any further clarification and needs.                                             

Best regards

Round 2

Reviewer 1 Report

Dear Author

Thanks for the revisions. The manuscript still requires english editing with native english speaker.